# Intelligent Sensor-Cloud in Fog Computer: A Novel Hierarchical Data Job Scheduling Strategy

**DOI:** 10.3390/s19235083

**Published:** 2019-11-21

**Authors:** Zeyu Sun, Chuanfeng Li, Lili Wei, Zhixian Li, Zhiyu Min, Guozeng Zhao

**Affiliations:** 1School of Computer Science and Information Engineering, Luoyang Institute of Science and Technology, Luoyang 471023, China; lylgszy@163.com (Z.S.); nywily@163.com (L.W.); zxlee0402@163.com (Z.L.); mindayu@163.com (Z.M.); ly_zgz@163.com (G.Z.); 2Key Laboratory of Intelligent IoT, Luoyang Institute of Science and Technology, Luoyang 471023, China

**Keywords:** wireless sensor networks, job scheduling, sensor cloud, scheduling strategy

## Abstract

In the Fog Computer (FC), the process of data is prone to problems such as low data similarity and poor data tolerance. This paper proposes a hierarchical data job scheduling strategy Based on Intelligent Sensor-Cloud in Fog Computer (HDJS). HDJS dynamically adjusts the priority of the job to avoid job starvation and maximize the use of resources, uses the key frame to the resource occupied information, distributes the frame sequence to the unit, and then combines the intra frame distribution strategy to balance the load between the nodes. The experimental results show our proposed strategy may be possible to avoid the operation of hunger and resource fragmentation problems, make full use of the advantages of multi-core and multi-thread, improve system resource utilization, and shorten the execution time and response time.

## 1. Introduction

The job-scheduling of the traditional wireless sensor network usually emphasizes a certain aspect but balances the property of the system from the whole view [1,2,3]. In addition to this, the division strategy of average node job makes the jobs between the nodes imbalanced, which leads to the problem that the nodes which are complicated in the task are put off the whole time of the job. Facing the increase in wireless sensor network data combined with the features of the job, developing a high speed job whose scheduling cost is low and that can also satisfy the users’ QoS needs is the key to improve the concurrent rate and to reduce the waiting time of user jobs. The distribution strategy of the average task is meaningful for reducing the influence of the finishing time of the nodes, wooden pail effect, and for improving the data computing ability [4,5,6]. Because the job scheduling and division strategy are meaningful for the improvement of the system property, scholars from all over the world currently study this area widely in relation to Wireless Sensor Networks (WSNs).

In WSNs, the most usual scheduling strategy, First Come First Service (FCFS), is based on the order of the jobs to schedule or to divide at the source [7]. The serious FCFS scheduling mechanism guarantees the fairness of the users but sacrifices the throughput efficiency and the source using rate of the system, which easily presents the problem of node free and node congestion [8,9,10]. The strategy of FirstFit based on the order that the jobs get into the queue of the jobs and run the first job whose source first satisfies others. Although the strategy can improve the throughput efficiency of the system dramatically, but the job whose source requirement is low can put off the job whose source requirement is high, which makes it not be solved for a long period [11]. The aspect of the fairness is limited, and it can also prolong the average waiting time of the system. Although the Job Scheduling Level Method (JSLM) has high throughput efficiency, it can prolong the time of calculating complicated jobs and also cannot satisfy fairness. BestFit strategy runs the job whose source is the highest and which can satisfy its requirement. At the same time, it can also lead to the phenomenon of job starvation [12,13].

In the WSNs based on sensor-cloud, with the rapid development of the high-performance computing and digital multimedia, the data render job scheduling becomes an important means to solve the single computing node [14,15]. The application has both computational and data intensive hybrid features [16]. At the same time, the emergence of big data technology and cloud computing technology further enhances the WSNs ability to render the extension.

The extension of WSNs computing scale has a high processing capacity, but if there is no system optimization strategy, the system cannot give full play to the computing power [17,18,19]. How to optimize the task scheduling strategy, assign the task to the nodes according to the computation, and improve the resource utilization and system performance is an important challenge for performance optimization research [20,21,22,23]. In addition, the expansion of the scale of the WSNs also means the increase of the energy consumption of the system [24,25,26]. How to design a reasonable scheduling strategy to reduce the energy cost of the system is also a very challenging problem in the WSNs.

The rest of this paper is organized as follows. Section 2 discusses the scheduling level method of job. The hierarchical job scheduling strategy was proposed in Section 3. Section 4 shows the simulation’s experimental results, and Section 5 summarizes the full paper.

## 2. Related Works

In the WSNs, when optimizing energy consumption, the property also plays an important role as a target. Aiming at the question which is about the balance of energy consumption and the property, there are many researchers currently working on the topic. He et al. [27] reduced the consumption of energy through minimizing the number of activated physical devices. Promoting a resource management framework, which is oriented to the property and the energy, Liu et al. [28] achieved the balance of property and energy by improving the knowledge of the theory. Aiming at the issue about the decreasing of the user service quality, which was caused by excessive integration of the resource, Wang et al. [29] solve the average optimization between the energy and the property by using the dynamic migration of the virtual machine. According to the platform of the SaaS, Zhao et al. [30] proposed a sort of resource scheduling scheme to balance the energy consumption and the property. The technology of Job Scheduling Optimization Strategy (JSOS) appears as a new technology in the current low energy consumption design of the computer system, which leads to many new challenges and chances for the energy-saving scheduling of the progress. Using the feature of JSOS technology to dispatch the task can make the high-performance cluster decrease the cost of energy the most in the whole system, considering the task performance. Sun et al. [31] proposed Heterogeneity-aware Loads Balancing (HLB) technology to balance distribution of the different tasks, which decreases the energy consumption overall, and to set the constraint of the temperature. Based on the Task Distribution Layer Method (TDLM) technology, Wang et al. [32] promoted a sort of heuristic scheduling algorithm to reduce the executive energy consumption, which was produced by the parallel task in the cluster environment. On the condition that this kind of algorithm does not influence the time of the whole tasks, the algorithm reduces the consumption of the energy by reducing the node voltage of the task. Liu [33] also did the same research to put forward a kind of task scheduling algorithm based on the energy-aware in the integrated environment, which reduces the power consumption through controlling the proper level of the voltage. Aiming to the JSLM proposed a sort of task scheduling strategy of the energy-aware, which adjusts the needed voltage dynamically, according to dividing the length of the subtask and minimizing the energy consumption under the condition of satisfying the user’s deadline. Wu et al. [34] promoted a sort of the energy saving system of the energy-aware, which judges whether the users are in the situation of activation or not by setting the dynamic reconfiguration models. Through turning on or turning off the application selectively to achieve the purpose of energy saving, this kind of system can acquire a good energy use rate, and at the same time, it can also provide a different energy scheduling strategy. Aiming to solve the task scheduling problem in the heterogeneous computing environment, Liu et al. [35] put forward an Energy-Conscious Scheduling (ECS) algorithm, which balances the task property and energy consumption by minimizing the value of Relative Superiority (RS) by making the RS of the task property and the cost of the energy consumption the optimization goal. On the basis of the optimization goal of the RS value, Sun et al. [36] combined the algorithm of the genes and proposed a meta-heuristic scheduling algorithm to optimize the property and the energy consumption of the tasks. Wang et al. [37] makes the energy consumption the constraint condition and makes the properties of the concurrent tasks the optimizing goals. Sun et al. [38] put forward a scheduling framework of the heterogeneous energy awareness resource, which makes the heat balance get rid of the ‘heat point’ by adjusting the voltage of the services dynamically. Li et al. [39] put forward energy awareness task fusion technology to optimize the energy consumption of the cluster, which reduces the energy consumption of the system through maximizing the method of using resources. Liu et al. [40] achieves the goal of optimizing the energy consumption by optimizing the free energy consumption and the running energy consumption effectively in the process of running the task. According to the parallel task scheduling, Shao et al. [41] put forward an optimizing algorithm, which concerns the running property, the cost, the reliability, even the energy consumption, and so on. However, this algorithm needs much exploration of the features of the task and even the features of the RS, and in the aspect of optimizing the energy, it also has much knowledge to explore.

In order to guarantee the fairness of the users’ work, the promoted work scheduling level method sets the dynamic priority for the work to guarantee the work priority scheduling method whose priority is high. When the source is limited, in the situation of not delaying the running of the original high priority work, the work priority of adjusting high priority dynamically will dispatch the work whose priority is low in advance, and at the same time, it will save resources for the work whose original priority is high [42]. Once the work requirement whose source satisfies the high level priority stopped the work whose priority is low and restarted to patch the work whose priority is high, the mission will send the frame sequence to the unit by way of making the unit the source particle size basic unit based on scene geometry, to choose the key frame. This uses the feedback of the source occupying information of the key frame and the related feature between the frames, combining the distributing strategy among the frames. The innovative points are as follows:(1)Setting the dynamic priority to guarantee the scheduling fairness.(2)Allowing the work whose priority is low and whose source requirement is small to run in advance to guarantee the adequate usage of the source, which also preserves the source for the work whose priority is high or whose requirement of source is big, and which also prevents situations of work starvation and reduction of source fragments.(3)The mission will send the frame sequence to the unit by way of making the unit the source particle size basic unit, based on scene geometry to choose the key frame, using the feedback of the source occupying information of the key frame and the related feature of between the frames. It combines the distributing strategy among the frames to render the work load between the nodes in balance.

## 3. Job Scheduling Level Method

### 3.1. Basic Concept

In the WSNs, the job scheduling layer mainly deals with the task submitted by the user. According to the needs of the users, the job is scheduled to the corresponding resource [43,44,45]. In order to better describe the method of job scheduling layer, the following definitions are introduced.

**Definition** **1.**
*In the queue waiting for the scheduling job, the fill-in job is the job that, at the back of the queue, is scheduled first.*


**Definition** **2.**
*The unit is a collection of CPU cores in the node.*


**Definition** **3.**
*The use rate shows the extent to which the sources in the system are busy. In general, we use the percentage of using source. However, it uses the percentage of the CPU usage to show the source’s use rate in the paper.*


The use rate of the system includes the real-time source use rate and the average source use rate. We assume that the real-time source use rate is *P_r_*, and the average source use rate is *P_r_T_* in the time period *T*. So that the real-time source use rate is the following:*P_r_* = *N_b_*/*N*(1)

In the Formula, *N_b_* indicates the amount of the CPU that is working; *N* shows the whole number of the CPU.

From the time period of *T_a_* to *T_b_*, the average source use rate is the following:(2)$$P_{r\_T}=\frac{1}{T_{a}-T_{b}}\int_{T_{a}}^{T_{b}}P_{r}(t)dt$$

Dividing this period of time into *m* areas, the time in every area is Δ*t*, and the result is as follows:(3)$$\Delta t=\frac{T_{a}-T_{b}}{m}$$

By using the conception of infinite segmentation, when *n*→*∞*, Δ*t*→0, the average source rate can be shown as:
(4)$$P_{r\_T}=\frac{1}{m}\underset{\Delta t\to 0}{\lim}\sum_{i=1}^{m}P_{r}(i\cdot \Delta t)$$

So that the average real-time source use rate in this paper states the source use rate in a certain time approximately.

Assuming a certain work *J_i_* whose original time of moment is *TJI*(*i*) is the moment that the work *J_i_* was submitted to the system by the users. Assuming that the start moment of the work is *TJS*(*i, k*), deadline moment is *TJF*(*i, k*), *k* is the number of the CPU that is needed. Therefore, the time of the work *J_i_* which needs *k* CPU is:(5)TJE(i,k)=TJF(i,k)−TJS(i,k)

**Definition** **4.**
*The whole time of the work indicates the whole finishing time of all the work in the work queue. It can also see it as the gap between the latest moment and the earliest moment.*


Therefore, the whole finishing time *TAJF*(*n*) of the *n* works in the work queue is:(6)TAJF(n)=MAX(TJF(i,k))−MIN(TJI(i))

**Definition** **5.**
*The response time of the work is the gap of the time between the original and the first time it starts, and it is also the waiting time of the work.*


**Definition** **6.**
*The turnover time of the work is the time gap between the start moment and the over moment. It not only includes the running time but the response time as well.*


Assuming that *TJres*(*i,k*) shows the response time of *k* CPU source works *J_i_*, and *TJave*_*res*(*n*) shows the average response time of *n* works, we can conclude that:(7)TJave(i,k)=TJS(i,k)−TJI(i,k)
(8)$$TJave\_res(n)=\frac{1}{n}\sum_{i=1}^{n}TJave\_res(i)$$

Assuming that *TJturn*(*i,k*) is the finishing time of *k* CPU source work *J_i_*, and *TJave*_*turn*(*n*) is the average turnover time of *n* works, we can conclude that:(9)TJturn(i,k)=TJF(i,k)−TJI(i,k)
(10)$$TJave\_turn(n)=\frac{1}{n}\sum_{i=1}^{n}TJave\_turn(i)$$

The analysis of the characteristics of operation is a sequence of frames, a group of mutual dependence and parallel split, in the WSNs [46]. Therefore, this method can be adjusted by dynamic priority and preemption mechanism to ensure fairness. The idea is to ensure maximizing the utilization of resources so that the free resources can be low priority and resources need little homework ahead of time, at the same time that resources are set aside for high priority and resource demanding work, in the WSNs.

### 3.2. Constraint Conditions for Data Jobs

Due to it being a data intensive application in terms of the memory requirements of the distribution of a wide range, there are dozens of MB of memory occupied by a simple scene, and it also has more than a dozen GB of memory of complex scenes [47,48]. Therefore, it is necessary to take account of the memory footprint in relation to the unit’s size. If you run a unit through all nodes of the process of the physical memory, the operating system will use the swap partition to ensure the normal operation of the process; so frequently, the disk data exchange will bring an extra-heavy overhead, and it will greatly extend the time. Therefore, the unit partition strategy needs to satisfy the following constraints:
(1)The memory of all the processes running on the unit does not exceed the physical memory capacity of the node,
(11)$$\sum_{i=1}^{p}m_{i}\leq M-\sigma$$
in which mi is the memory for each process, M is the machine memory capacity, p is the number of processes running simultaneously, and σ is the memory of the system.In addition, it is also a computationally intensive application processor; task switching overhead will usually offset the boost processor sharing performance and even lead to performance decline. Therefore, the unit partition strategy should satisfy the following constraints:(2)The sum of the number of threads that run on the unit cannot exceed the number of processors in the node,
(12)$$\sum_{i=1}^{p}t_{i}\leq C$$
in which ti is the number of threads in the i process, C is the number of processor cores for the node, and p is the number of simultaneous processes.

According to the above two constraints, the switching cost and the lack of memory can be eliminated. Therefore, to maximize the use of resources, the goal of the division of the unit needs to meet the above two constraints to start the concurrent process to reduce the time as much as possible. Assume that the optimal number of units in each node is *C_o_*, the number of units corresponding to *N_u_*, then the optimal process number *P_o_* and the number of threads allocated by each process can be given as
(13)$$\left\{ \begin{array}{l} N_{u}=P_{o}\\ P_{o}=\max\{p|pm_{i}\leq M-\sigma ,C\%p=0\}\\ T_{o}=C_{o}\\ C_{o}=\frac{C}{P_{o}} \end{array}\right.$$

According to the above analysis, it can be seen that the unit partition strategy first selects the corresponding resource for the user according to the parameters of the task submitted by the user. Here the use of CPU as a resource for the number of units and then the method of scene geometry selection based on key frames and key frames from the pre-rendered are used to estimate resource usage according to the resource of that unit optimal kernel number, and each CPU is assigned a number of processes, and finally, assigned tasks according to a certain strategy in the WSNs.

After submitting the task, the schedule that every node finishes all tasks is:(14)$$\begin{array}{l} T_{1}:\{t_{1},t_{2},\ldots ,t_{r-1}\},\\ T_{2}:\{t_{r},t_{r+1},\ldots ,t_{2r-1}\},\\ \;\vdots \;\;\;\;\;\vdots \\ T_{n}:\{t_{nr},t_{nr+1},\ldots ,t_{m}\} \end{array}$$

Giving the example of the node *i*, the paper describes the energy consumption of the single node. Assuming that *T_i_* is the whole time of the node *i, it* is as follows:(15)$$T_{i}=\sum_{j=ir}^{(i+1)r-1}t_{j}$$

Therefore, the finishing time *T*_max_ of the whole nodes are the latest nodes. The accurate calculating process is as follows:(16)$$T_{\max}=\max(T_{1},T_{2},\ldots ,T_{n})$$

In order to illustrate the energy consumption *E_i_* of the node, which is number *i*, Equation (17) illustrates the process of the calculation.
(17)$$E_{i}=W_{i}+I_{i}$$

In the Equation (17), *W_i_* is the whole energy consumption of the node *i* which finishes the task list L:{*l_ir_*,*l_ir_*_+1_,…,*l*_(*i*+1)*r*-1_}. This part of energy consumption is decided by the power *P_i_* of the node *i* and the running time of the task. *I_i_* shows the energy consumption of the node *i* in this scheduling, which is in the free situation; this part of the energy consumption is decided by the free power *p_i_* of the node *i* and the free waiting time. This paper assumes that if and only if the nodes finish the whole task in the queue, they can be in the situation which is free. The Equations (18) and (19) describe two kinds of counting process of the energy consumption which are above.
(18)$$E_{i}=W_{i}+I_{i}$$
(19)$$I_{i}=p_{i}(T_{\max}-T_{i})$$

Then, if we put the Equations (8) and (9) into (10), then we can conclude the energy consumption of the number *i* node:(20)$$E_{i}=\sum_{j=ir}^{(i+1)r-1}P_{i}t_{j}+p_{i}(T_{\max}-T_{i})$$

According to the energy consumption model of the node *i*, in the nodes list (*R*_1_,*R*_2_,…,*R*_n_), the Equation (21) describes the counting process of the whole nodes’ energy consumption.
(21)$$E=\sum_{i=1}^{n}\sum_{j=1}^{m}P_{i}t_{j}+\sum_{i=1}^{n}p_{i}(T_{\max}-T_{i})$$

In the WSNs, the whole energy consumption includes the energy consumption of all tasks and time of node and even the free waiting energy consumption. If Equation (15) is put into Equation (20), eventually we can get the energy consumption computing model.

The accurate estimation of resource occupancy is the premise of the efficient implementation of the multi-process mode, and how to divide the granularity of computing resources based on the feedback resource information is the key to improve the performance of multi-process [49,50]. The traditional way of dividing the resources of the unit is to improve the performance of the node; dividing it into coarse granularity and multi-thread is not obvious, therefore, this paper puts forward the concept of "unit" as the resource partition unit.

Multi-frame sequences are usually assigned to this set for multi thread, and the unit contains the number of cores that show how many threads are rendered. In this paper, we assume that a kernel corresponds to a thread, and each sub job (4 frames) is rendered in the order of the number of frames. Therefore, for nodes with multi-core processors, according to the resource information feedback, the number of nuclear units is the optimal unit allocation by starting a process to render sequence frame multi-thread allocation and resource utilization and by maximizing the increased efficiency.

In order to eliminate the influence of the accumulation of inter frame differences on the load balance among the nodes, the allocation strategy uses the correlation between the frame and the frame to allocate the frame to the unit. In the traditional way, the continuous frames are assigned to the nodes sequentially, and the correlation between frames can only guarantee the difference between adjacent frames. However, if both ends of the sequence, frame node 1 and node *R*, are assigned to the total operation in the sequence frame, this will bring a larger load imbalance; especially in the abrupt shot explosion, some background conversion, the unbalanced phenomenon is more serious. Therefore, in the unit distribution frame, according to the frame number, staggered modes are allocated for each node frame number of successive frames will be assigned to the unit adjacent to the purpose is to use inter frame correlation to minimize the load of each unit. Therefore, the unit *r* is assigned to the frame sequence set *S_r_* can be given as
(22)$$S_{r}=\{f|f=Rz+r,z\in N,1\leq f\leq F\}$$

The frame allocation strategy based on unit belongs to a static strategy, while dynamic strategy needs to start the new process and reload the additional cost of the scene file. The implementation of this strategy is relatively simple; there is no preprocessing or task migration overhead.

### 3.3. Data Job Scheduling

The user submits 5 jobs to the system. According to the job scheduling strategy proposed in this paper, the job scheduling process is shown in Figure 1.

Figure 1a shows the initial state of the scheduling, the current system has idle resources, the user submitted to the 5 jobs, *J*_1_–*J*_5_, waiting for job management system scheduling. *J*_1_, *J*_3_, and *J*_5_ have the highest priority, priority scheduling; the current system has 15 idle states of CPU, to meet the needs of the resources of these three operations. As a result, *T*_0_, *J*_1_, *J*_3_, and *J*_5_ are assigned resources to perform. Since the current system does not have an idle resource, it cannot meet the resource requirements of job *J*_2_ and *J*_4_, so job *J*_2_ and *J*_4_ remain in the waiting queue.

Figure 1b represents that time T1 is the scheduling state. At this time, Job *J*_5_ has finished the rendering, and there are 4 idle CPUs. According to the order of priority, *J*_2_ should be scheduled and allocated resources. However, *J*_2_ requires 8 CPUS, which cannot be met by the system. For the job *J*_4_ with lower priority, its resource requirement can be met by the system. Therefore, in order to maximize the throughput and the resource efficiency, the job scheduling strategy schedules *J*_4_ in advance. Henceforth, the job *J*_4_ with lower priority is executed before *J*_2_ with higher priority. 

Figure 1c represents that time *T*_2_ is the scheduling state. At this time, the data fusion for *J*_3_ is finished, and the number of idle CPUs is 5 for the system. Therefore, the resource requirement of *J*_2_ still fails to be satisfied. However, the number of appointed CPUs is 4, and the total number of idle CPUs and appointed CPUs is 9, which could satisfy the requirement of *J*_2_. Meanwhile, the progress of *J*_4_ is smaller than 90%. Therefore, *J*_4_ is paused, and the CPUs occupied by *J*_4_ are now changed into idle state. Then, 8 idle CPUs are allocated to *J*_2_, and *J*_4_ is now in the queue waiting to be scheduled. At this time, since all the jobs in the queue require more than 1 CPU, the system will generate 1 idle CPU. 

Figure 1d represents that time *T*_3_ is the scheduling state. At this time, the data fusion task for *J*_2_ is finished, and there are 9 idle CPUs in the system, which could satisfy the requirement of *J*_4_. Then, resources are allocated to *J*_4_ for the unfinished jobs. At this time, the scheduling queue is empty. The sensor-cloud computation system waits for users to submit jobs.

In order to compare the scheduling process and the problems of the strategy, the scheduling results of FCFS and FirstFit job scheduling policies for the 5 jobs are shown in Figure 2. FCFS scheduling strategy of operating is in strict accordance with the arrival order (priority) scheduling; the resources are idle, and due to the blocking phenomenon in the job scheduling process, job completion time is longer, and the system resource utilization rate is not high. The scheduling results are shown in Figure 2a. FirstFit job scheduling policy can also generate resource idling and also cannot guarantee fairness; scheduling results are shown in Figure 2b.

When the job scheduling layer assigns the resource to the task in the WSNs, the task distribution layer decomposes the job into a task and distributes it evenly to the resource. The task distribution layer method is divided into 3 steps: (1)First of all, it is based on the key frame before rendering the granularity of the resources.(2)Secondly, the correlation based on frame will render the frame sequence assigned to the unit.(3)Finally, combined with intra partition strategy, it will not be divisible by the number of the remaining frames unit assigned to the unit.

Aiming at the problems such as low data similarity and poor data tolerance, this paper proposes a hierarchical job scheduling strategy (HDJS). The implementation of the proposed HDJS is as follows (Algorithm 1):

**Algorithm 1** Hierarchical Job Scheduling Strategy**Input**: Wait for scheduling job queue**Output**: Job scheduling sequence1. Set flag = 0;2. Sort jobs in SJobs by priority from high to low;3. for each *J_i_* in SJobs do4.   if NumJ_i_ <= freeNum then5.      Allocate the free CPUs to *J_i_*;6.      Update the state of CPUs and Rjobs;7      if flag ==1 then8.        Set *J*_i_ as fill-in job and corresponding CPUs as reservation CPUs;9.        Update order *N*;10.     endif11. break12.  else13.     if Num*J_i_* > freeN && Num*J_i_* <= (free*N* + order*N*) then14.        Select jobs paused in IJobs and put them to bJobs;15.        Select CPUs corresponding to jobs in bJobs and put them to bCs;16.        Update bCsN;17.     if bCs ! = NULL then19.        Change the state of CPUs to free and CPUs as no reservation CPUs;20.        Update free*N*, order*N*;21.    endif22.    endif22.     if Num*J_i_* <= free*N* then23.        Allocate the free CPUs to *J_i_* and update the state of CPUs;24.        Update busy*N*;25.     if flag == 1 then26.        Set *J_i_* as fill-in job and corresponding CPUs as reservation CPUs;27.        Update order*N*;28.     29.     endif30.     endif33. flag = 1;34. endfor

The Algorithm 1 is used to judge whether the current job initialization is a sign of queue operations (flag) algorithm simulation having completed the process of scheduling a period in the WSNs. If the current idle state of the CPU resource is able to meet the resource requirements of the job *J_i_*, the idle state CPU is assigned to the job *J_i_*, and then the resource state and the busy state CPU number are updated. If the next one and the reservation status of homework then set the *J_i_* queue operations, and the corresponding CPU is set to CPU resources reservation number at the same time, update the appointment booking status CPU. The time complexity of the algorithm is O(*nlogn*), and the time is mainly consumed in the job priority ranking of the waiting list. In this paper, we use a stable merge sort, and the space complexity is O(*n*).

If frames cannot be divisible by the number of nodes, there are some nodes in idle state when there is a residual frame; resources cannot be fully utilized, and finishing time will be a prolonged operation. Fine grained intra frame distribution is helpful to balance the load of each node, but the efficiency is not high. As a result, an intra-frame distribution strategy is used to render a smaller residual frame, to avoid the idle resources, to further balance the load between nodes, and to reduce the completion time of the job.

By assigning the fine-grained task to the unit, the number of blocks per frame *N*-tile can be given as:(23)$$N_{tile}=\left\lfloor \frac{R}{\left((F-R)\;\mathrm{mod}R\right)}\right\rfloor$$

## 4. Experimental Results and Analysis

In order to evaluate the proposed scheduling strategy performance in the WSNs, this paper builds WSNs based on B/S architecture using local server. In the simulation of the WSNs, the management node undertakes the management job and resource management functions. The system deployed on the management node is the Tomcat version 6.02; the version of the database is mysql-5.32. In addition, the 200 nodes responsible for the hardware configuration of each node are the same, belonging to the isomorphic WSNs. 

First, Systems test the resource utilization, and the priority of each task, and the proportion of resource requirements is different. Figure 3 shows the comparison of resource utilization ratio for different strategy. It is clearly evident from Figure 3 that our proposed HDJS has better performance and higher resource utilization. When the number of jobs is 100, the resource utilization ratio of HDJS is increased by an average of 8.8% compared with FCFS, and the best case is increased by 10.28%. When the number of jobs is 200, the resource utilization ratio of HDJS is increased by an average of 8.51% compared with FCFS, and the best case is increased by 10.02%. With the increase of job resources and high priority, the HDJS can obtain higher resource utilization ratio in the WSNs.

Figure 4 shows the comparison of average completion times for different strategies. As shown in Figure 4, when the number of jobs is 100, the average execution time of HDJS is reduced compared with FCFS average execution time. When the amount of jobs is 200, the average execution time of HDJS is reduced compared with FCFS average execution time. Although FirstFit is based on the principle of matching the system resources, the task can be scheduled as soon as possible so as to avoid the resource congestion caused by the lack of system resources. However, in the process of matching resources, FirstFit will produce the phenomenon of idle resources, which cannot make full use of the resources. HDJS can achieve the operations and does not make the resources idle in idle waiting state, while the low priority and high degree of idle resources work to fill in this part for resources, not only to improve their utilization rate but also to make sure the operations can complete part of the schedule. In this way, each job has the opportunity to run ahead of time not because of low priority and long wait for the scheduling state. Of course, this part of the low priority and early scheduling of the work in the system resources to meet the needs of high priority resources will automatically release the resources, also to meet the principle of fairness. This kind of advance plug in execution can reduce the average turnaround time in the WSNs.

Figure 5 shows the comparison of data delay ration for a different strategy. It is clearly evident from Figure 5 that our proposed HDJS is superior to the other two algorithms in data delay ration. The user experience is not only the number of hours of work but also includes the fairness factor. Usually, the user’s needs are not only related to the resources but also to the fairness of their request. For users, fairness is a very important factor. Users usually, from their own point of view, do not want to submit their own work whose execution has been delayed, which is the key point to render the user needs to deal with WSNs. The total completion time is one of the key indicators to evaluate the performance of the system. Compared with the traditional job scheduling strategies FCFS and FirstFit, HDJS can achieve higher performance and better user experience in WSNs management in the WSNs. The traditional FCFS job scheduling strategy can guarantee absolute fairness, which is strictly based on the order of the user submitted jobs. However, the algorithm does not consider the improvement of system performance. FirstFit scheduling strategy with respect to the FCFS strategy readjusts the scheduling queue in order to achieve the purpose of increasing the throughput of the system, but its consideration in terms of fairness is poor; this is often not acceptable for users. In this paper, job scheduling strategy, according to the job priority scheduling, will break absolute fairness only when the job is blocked. With high priority, resource needs work temporarily to wait; when the priority is low, resource needs little homework ahead of time scheduling, thus speeding up the process of scheduling, reduce the resource idle time, and make full use of the current free system resources. For the scheduling jobs in the current system, the dynamic priority policy can improve the throughput of the system, and the reservation mechanism ensures the fairness of job scheduling with high priority and large resource requirements. Therefore, the method of this paper can shorten the total completion time of the work and can provide the overall performance of the system.

Figure 6 shows the sensor-cloud processing time of the different testing cases in the low cyber conditions. From Figure 6a,b, we can conclude that HDJS has the superior property, compared with other two optimizing approaches. In the best situation, the property of the HDJS increased by 13%, compared with HLB, and increased by 7%, compared with JSOS. Figure 6c,d shows the localization execution rate of the different testing cases in WSNs period. From Figure 6, it can be concluded that the HDJS has the highest localization execution rate, compared with other two kinds of approaches. For example, when the data block is 64 KB, the localization execution rate of the HDJS can increase to 23.7%; however, the localization execution rates of the HLB and JSOS are only 12.28% and 15.51%, respectively. Compared with the HLB, the property of the HDJS and localization execution rate of it increased dramatically. This is because, in the approach of the HLB, the data blocks were put on the working nodes randomly according to the hard disk of the nodes using rate. This strategy does not concern the differences of the working nodes calculating in WSNs, which caused that, after the nodes whose calculating abilities finish the local mission stole the local sensor-cloud tasks of the working nodes which have low computing abilities, which leads to abundant non-localization execution of the task sensor-cloud. The non-localization execution of the sensor-cloud task caused the distant communication of the nodes in WSNs, so that it can prolong the time that the jobs stay in WSNs. The property of the HDJS and the localization execution rate of the sensor-cloud have improved to some extent, compared with JSOS. This is because when the JSOS starts to select the input data nodes, according to the calculating rate of different working nodes, it preserves the proper amount of the data block, and the nodes whose calculating ability is high preserve a bigger number of the sensor-cloud task inputting data blocks, which reduced the number of the data blocks movement. However, in the approach of JSOS, the execution time of the task has a linear relation with the amount of the data that is inputted. 

According to the analysis in the third chapter, the execution time of the task has no simple linear relationship with the amount of data that is imputed. It also has some relationship with the features of the loads and the features of the working nodes. Therefore, the JSOS approach can also lead to that part of the working nodes to execute the non-localization sensor-cloud task. The HDJS proposed in the paper uses the property forecasting model to test the process abilities of the working nodes, whose forecast accuracy is higher than JSOS. In addition to this, in the period of sensor-cloud, the HDJS can dynamically monitor different working nodes in the local sensor-cloud task list execution process. Because of this, it reduces the waiting cost of the inputting data node-hop cyber communication that was caused by the non-localization sensor-cloud mission in the WSNs.

## 5. Conclusions

In big data environments and WSNs, data plays an important role in improving system performance and job scheduling. With the strategy proposed in this paper, it may be possible to avoid the operation of hunger and resource fragmentation problems, make full use of the advantages of multi-core and multi-thread, and improve system resource utilization, considering the idle energy consumption of the system and the energy consumption of the task runtime; according to the natural parallelism of the rendering application frame and the frame, the rendering task energy consumption model is established. According to this model, the task scheduling queue is split into sub-queues, and the sub-sequence is optimized. Task scheduling improves node utilization; avoids waste of idle node energy consumption, thus completing the optimization of energy consumption of the rendering system’s global task; and shortens the execution time and response time. In future work, we will combine simulated annealing algorithm and GPGPU platform with the proposed strategy so as to reduce the system execution time more effectively.

## Figures and Tables

**Figure 1 sensors-19-05083-f001:**
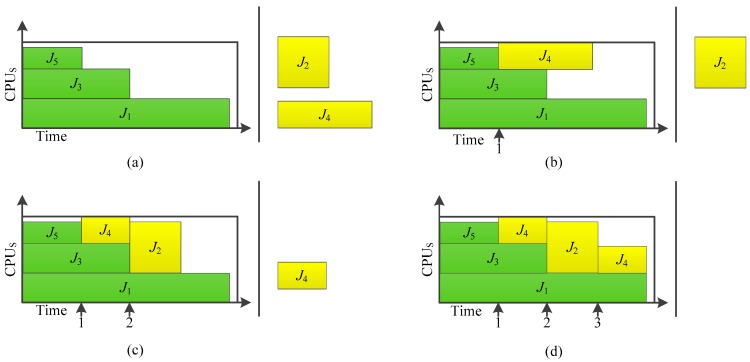
Job scheduling instance based on HDJS; (**a**) *t* = 10 s; (**b**) *t* = 20 s; (**c**) *t* = 30 s; (**d**) *t* = 40 s.

**Figure 2 sensors-19-05083-f002:**
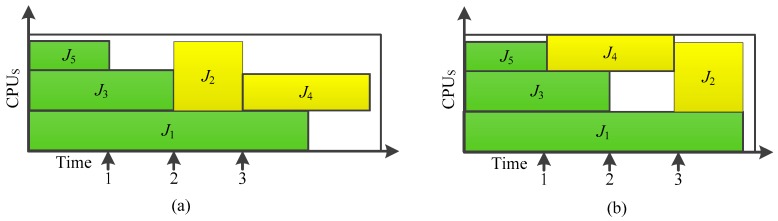
The process of FCFS and FirstFit job scheduling (**a**) *t* = 100 s; (**b**) *t* = 150 s.

**Figure 3 sensors-19-05083-f003:**
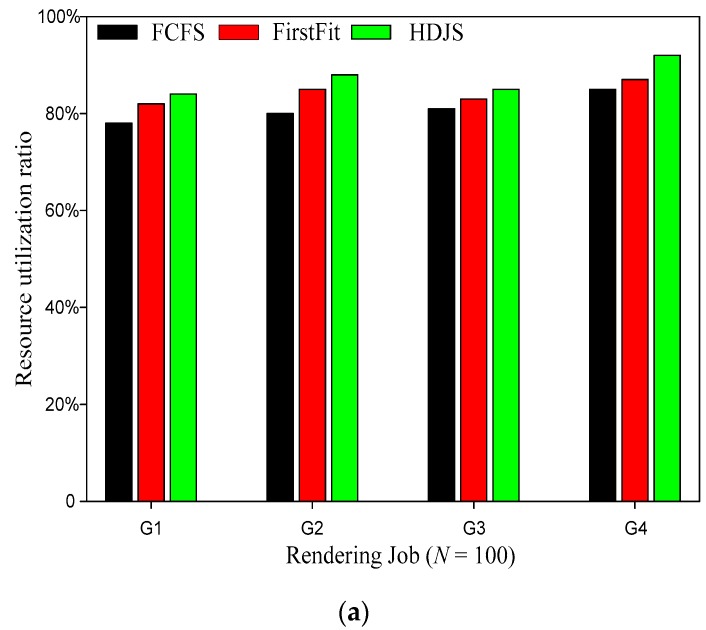
Comparison of resource utilization ratio for different strategies; (**a**) *N* = 100; (**b**) *N* = 200.

**Figure 4 sensors-19-05083-f004:**
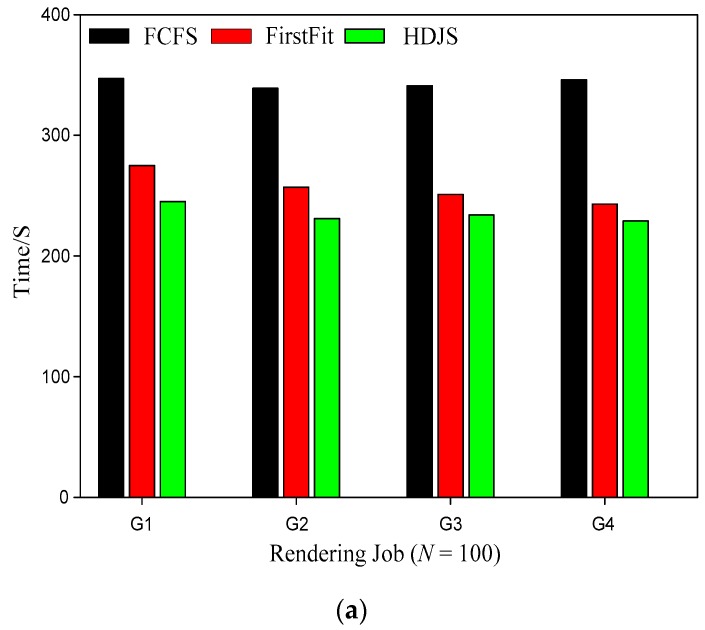
Comparison of average completion time for different strategies; (**a**) *N* = 100; (**b**) *N* = 200.

**Figure 5 sensors-19-05083-f005:**
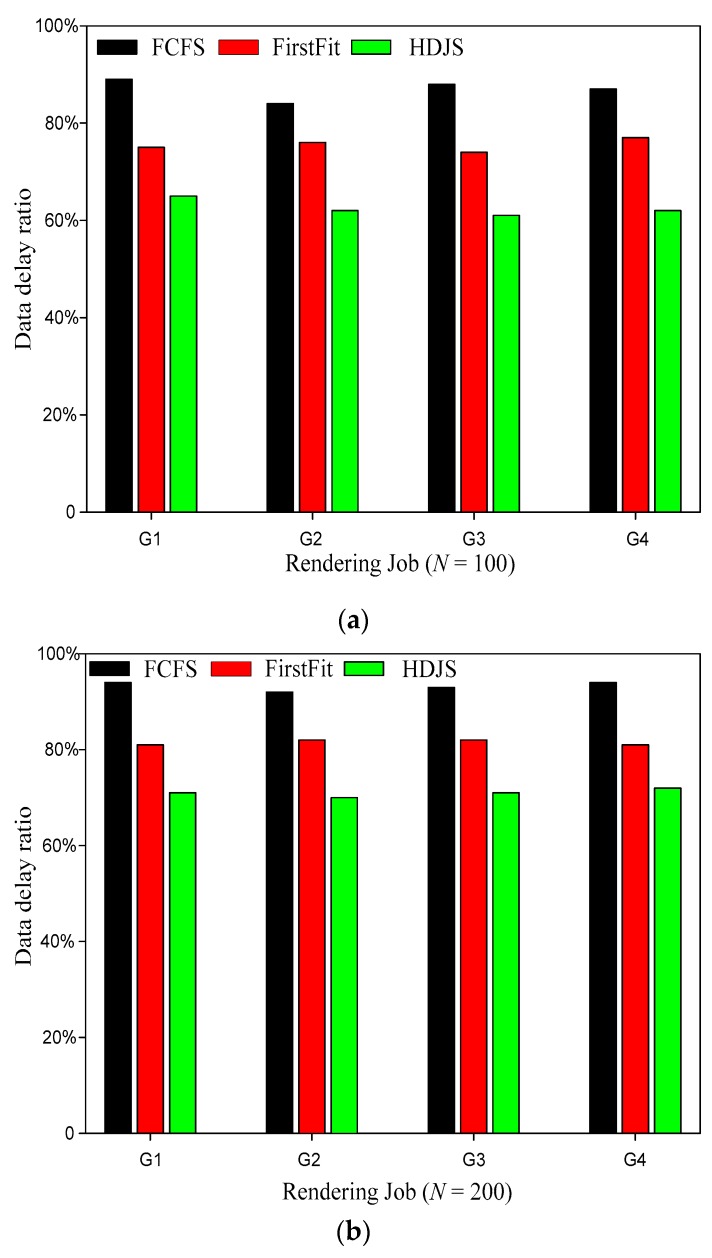
Comparison of data delay ration for different strategies; (**a**) *N* = 100; (**b**) *N* = 200.

**Figure 6 sensors-19-05083-f006:**
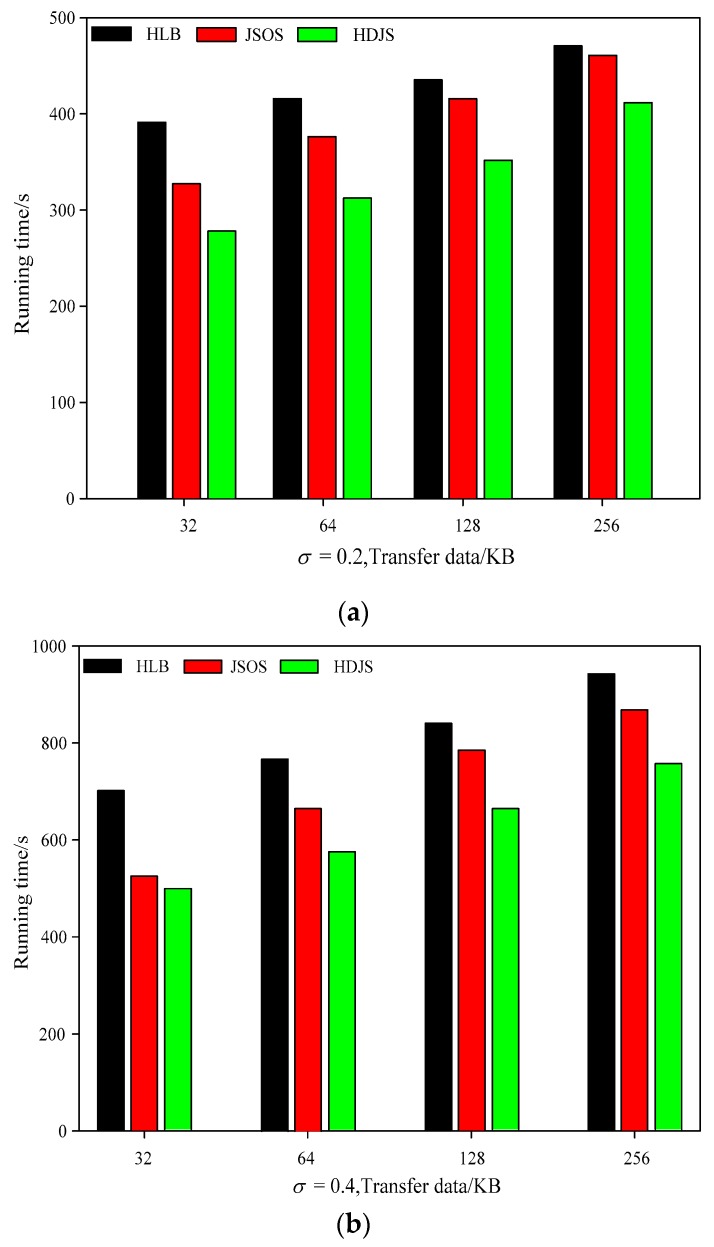
Comparison of data transmission and running time under different parameters; (**a**) *σ* = 0.2; (**b**) *σ* = 0.4; (**c**) *σ* = 0.6; (**d**) *σ* = 0.8.

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
