# Peer review of "Intelligent Sensor-Cloud in Fog Computer: A Novel Hierarchical Data Job Scheduling Strategy"

_sensors, 2019, doi:10.3390/s19235083_

Round 1

Reviewer 1 Report

In the section "Related Works" the uses the structure "Paper [X]" to cite the main contributions. I think that this type of quotation detracts a fair protagonism from the authors of the research that is cited, so I propose that a quotation be made in the same way like this one: "AutorName(s) [X] proposed...".

The paragraphs in the "Related Works" section are excessively long, which makes them difficult to read/understand. It is recommended that each paragraph group together the works that are connected to each other so that each paragraph can suggest to the reader a new idea or researching approach.

Limit the use of the first person plural "we" throughout the document.

At the end of the "Related Works" section, there is no clear link between the contributions of other authors and the work proposed below. There is a need for authors to better explain the contributions [41-45] separately. It is also necessary to establish a research objective at the end of this section that is a consequence of the shortcomings detected in the state of the art and that justifies the following sections.

The theoretical framework is well justified although the behaviour of the model with respect to the use of multi-core or multi-thread systems is not well explained. Why does this system behave better than others when multi-cores are used? I think that the use of multi-cores is not an inherent advantage of this method but is related to the parallelism of the requests/data.

Figure 1 should indicate that the HDJS strategy is being used.

The algorithm in line 314 should indent properly. (see 21. endfor)

The conclusions section is a mere repetition of the abstract. The main contributions of the work should be indicated. This section should answer the question: why should I choose HDJS over other strategies? The limitations of this strategy should also be included.

In addition, future work is not clearly explained, what type of AI-based algorithm is to be used?, what would be its advantages?

The authors do not talk about GPGPU platforms (massively multi-core), would it be appropriate to use them with this strategy?

Finally, a fully linguistic revision of the paper is recommended.

Author Response

In the section "Related Works" the uses the structure "Paper [X]" to cite the main contributions. I think that this type of quotation detracts a fair protagonism from the authors of the research that is cited, so I propose that a quotation be made in the same way like this one: "AutorName(s) [X] proposed...".

We would like to thank the reviewer for the helpful suggestion. I have modified the references as requested by the experts.

In the WSNs, when optimizing the consumption of the energy, the property also plays an important role as a target. Aiming at the question which is about the balance of the energy consumption and the property, there have been many researchers doing many researches currently. He, S et.al [26] reduced the consumption of the energy through minimizing the number of the activated physical devices. Promoting a resource management framework which is oriented to the property and the energy, Liu, Z et.al [27] achieved the balance of the property and the energy by improving the knowledge of the theory. Aiming at the issue about the decreasing of the user service quality which was caused by much integrations of the resource, Wang, T et.al [28] solves the average optimization between the energy and the property by using the dynamic migration of the virtual machine. According to the platform of the SaaS, Zhao, G.Z et.al [29] proposed a sort of resource scheduling scheme to balance the energy consumption and the property. The technology of Job Scheduling Optimization Strategy (JSOS) appears as a new technology in the current low energy consumption design of the computer system, which leads to many new challenges and chances for the energy-saving scheduling of the progress. Using the feature of JSOS technology to dispatch the task can make the high performance cluster decrease the furthest cost of the energy in the whole system, in the situation of the consideration to the task performance. Sun, Z.Y et.al [30] proposed Heterogeneity-aware Loads Balancing (HLB) technology to balance distribution of the different task, which decrease the energy consumption overall and to set the constraint of the temperature. Based on the Task Distribution Layer Method (TDLM) technology, Wang, T et.al [31] promoted a sort of heuristic scheduling algorithm to reduce the executive energy consumption which was produced by the parallel task in the cluster environment. On the condition that this kind of algorithm does not influence the time of the whole tasks, the algorithm reduces the consumption of the energy by reducing the node voltage of the task. Liu, X.X et.al [32] also did the same research to put forward a kind of task scheduling algorithm based on the energy-aware in the integrated environment, which reduces the power consumption through controlling the proper level of the voltage. Aiming to the JSLM proposed a sort of task scheduling strategy of the energy-aware, which adjusts the needed voltage dynamically according to dividing the length of the subtask and minimize the energy consumption in the condition of satisfying the user deadline. Wu, Y.K et.al [33] promoted a sort of the energy saving system of the energy-aware, which judges the users are in the situation of activation or not by setting the dynamic reconfiguration models. Through turning on or turning off the application selectively to achieve the purpose of energy saving, this kind of system can acquire a good energy using rate. And at the same time, it can also provide different energy scheduling strategy. Aiming to the task scheduling problem in the heterogeneous computing environment, Liu, X.X et al [34] put forward an Energy- Conscious Scheduling (ECS) algorithm, which balances the task property and energy consumption by minimizing the value of Relative Superiority (RS) according to making the RS of the task property and the cost of the energy consumption as the optimization goal. In the basis of the optimization goal of the RS value, Sun, Z.Y et.al [35], combined the algorithm of the genes, purposed a meta-heuristic scheduling algorithm to optimize the property and the energy consumption of the tasks. Wang, J.F et.al [36] makes the energy consumption as the constraint condition and makes the property of the concurrent tasks as the optimizing goals. Sun, Z.Y et.al [37] put forward a scheduling framework of the heterogeneous energy awareness resource, which makes the heat balance get rid of the ‘heat point’ according to adjusting the voltage of the services dynamically. Li, Z.X et.al [38] put forward energy awareness task fusion technology to optimize the energy consumption of the cluster, which reduces the energy consumption of the system through the way of maximizing the method of using resources. Liu, X.X et.al [39] achieves the goal of optimizing the energy consumption by optimizing the free energy consumption and the running energy consumption effectively in the process of running the task. According to the parallel task scheduling, Shao, Y.L et.al [40] put forward an optimizing algorithm which concerns about the running property, the cost, the reliability, even the energy consumption and so on. However this algorithm needs much exploration in the features of the task and even the features of the RS, and, in the aspect of optimizing the energy, it also has much knowledge to explore.

In order to guarantee the fairness of the users work, the promoted work scheduling level method sets the dynamic priority for the work to guarantee the work priority scheduling method whose priority is high. When the source is limited, in the situation of not delaying the running of the original high priority work, the work priority of adjusting high priority dynamically will dispatch the work whose priority is low in advance, and at the same time it will save resource for the work whose original priority is high [41]. Once the work requirement whose source satisfies the high level priority stopped the work whose priority is low and restarted to patch the work whose priority is high, the mission will send the frame sequence to the unit by the way of making the unit as the source particle size basic unit, basing on the way of scene geometry to choose the key frame, using the feedback of the source occupying information of the key frame and the related feature of between the frames, combining the distributing strategy among the frames. The innovative points are as follows:

1) Setting the dynamic priority to guarantee the scheduling fairness.

2) Allowing the work whose priority is low and whose source requirement is small to run in advance to guarantee the adequate usage of the source, which also preserves the source for the work whose priority is high or whose requirement of source is big, and it also prevents situations of work starvation and reducing the source fragments.

3) The mission will send the frame sequence to the unit by the way of making the unit as the source particle size basic unit, basing on the way of scene geometry to choose the key frame, using the feedback of the source occupying information of the key frame and the related feature of between the frames and it combined the distributing strategy among the frames to render the work load between the nodes in balance.

The paragraphs in the "Related Works" section are excessively long, which makes them difficult to read/understand. It is recommended that each paragraph group together the works that are connected to each other so that each paragraph can suggest to the reader a new idea or researching approach.

We would like to thank the reviewer for the helpful suggestion. I have already reduced the related work and highlighted the research method of the paper. (Ask the experts teacher to refer to the first question.)

Limit the use of the first person plural "we" throughout the document.

We would like to thank the reviewer for the helpful suggestion. I have made relevant changes.

At the end of the "Related Works" section, there is no clear link between the contributions of other authors and the work proposed below. There is a need for authors to better explain the contributions [41-45] separately. It is also necessary to establish a research objective at the end of this section that is a consequence of the shortcomings detected in the state of the art and that justifies the following sections.

We would like to thank the reviewer for the helpful suggestion. I have described [41]-[45] and gave the analysis process, and added the research purpose, and method in the last paragraph of the relate works. (Ask the experts teacher to refer to the first question.)

The theoretical framework is well justified although the behaviour of the model with respect to the use of multi-core or multi-thread systems is not well explained. Why does this system behave better than others when multi-cores are used? I think that the use of multi-cores is not an inherent advantage of this method but is related to the parallelism of the requests/data.

We would like to thank the reviewer for the helpful suggestion. I agree with you. Firstly, on the fog-computation resources, we order the cut-in-line jobs according to their priorities. Then we go through all the jobs in the cut-in-line job list. If the progress of a cut-in-line job is smaller than a threshold, we will calculate the overall CPUs occupied by these cut-in-line jobs. Otherwise, we switch to the next job. If the overall CPUs occupied by these cut-in-line jobs could satisfy the resource requirement for scheduling the current job, the exhaustive research will be stopped, and we pick up the corresponding cut-in-line job. After that, we compare the priority of the picked job and that of the current job. When the priority of the current job is higher than those of all the cut-in-line jobs, the execution of these cut-in-line jobs will be paused. Then these jobs will be stored in the pausing list.

Figure 1 should indicate that the HDJS strategy is being used.

We would like to thank the reviewer for the helpful suggestion. I have made relevant changes. Figure1. Job scheduling instance based on HDJS

The algorithm in line 314 should indent properly. (see 21. endfor)

We would like to thank the reviewer for the helpful suggestion. I have made relevant changes.

The conclusions section is a mere repetition of the abstract. The main contributions of the work should be indicated. This section should answer the question: why should I choose HDJS over other strategies? The limitations of this strategy should also be included. In addition, future work is not clearly explained, what type of AI-based algorithm is to be used?, what would be its advantages? The authors do not talk about GPGPU platforms (massively multi-core), would it be appropriate to use them with this strategy?

We would like to thank the reviewer for the helpful suggestion. I have revised the conclusion in accordance with requirement of the experts. At the same time, it adds a description of future works.

In big data environment and the WSNs, data plays an important role in improving system performance and job scheduling. In this paper, it proposed strategy may be possible to avoid the operation of hunger and resource fragmentation problems, make full use of the advantages of multi-core and multi thread, improve system resource utilization, considering the idle energy consumption of the system and the energy consumption of the task runtime, according to the natural parallelism of the rendering application frame and the frame, the rendering task energy consumption model is established. According to this model, the task scheduling queue is split into sub-queues, and the sub-sequence is optimized. Task scheduling to improve node utilization and avoid waste of idle node energy consumption, thus completing the optimization of the energy consumption of the rendering system global task, and shorten the execution time and response time. In future work, we will combine simulated annealing algorithm and GPGPU platform with the proposed strategy, so as to reduce the system execution time more effectively.

Finally, a fully linguistic revision of the paper is recommended.

We would like to thank the reviewer for the helpful suggestion. We have checked the paper for the correctness of the grammar and typos.

Reviewer 2 Report

Based on the sensor-cloud and Fog Computing, the authors proposed a Hierarchical Data Job Scheduling Strategy (HDJS). This strategy could fully exploit the limited resources in wireless sensor networks to aggregate the data chips into a data entity, and then employ fog computation for further data fusion. Finally, the effectiveness of the proposed algorithm is verified by simulation results.

Generally, this work exhibits certain novelty and feasibility. This paper is well organized with sufficient proof. The questions for the authors are listed as follows.

In Section 2, please organize the related works according to the timeline of the study on data fusion. In Section 3.1, how can the authors guarantee the contention fairness when the data in the sensor-cloud computation apply for the resources? The explanation on Fig. 1 is over-simplified. Please supplement more details. Please check the paper to avoid grammar mistakes and typos.

Author Response

In Section 2, please organize the related works according to the timeline of the study on data fusion.

We would like to thank the reviewer for the helpful suggestion. I have modified the references as requested by the experts.

In the WSNs, when optimizing the consumption of the energy, the property also plays an important role as a target. Aiming at the question which is about the balance of the energy consumption and the property, there have been many researchers doing many researches currently. He, S et.al [26] reduced the consumption of the energy through minimizing the number of the activated physical devices. Promoting a resource management framework which is oriented to the property and the energy, Liu, Z et.al [27] achieved the balance of the property and the energy by improving the knowledge of the theory. Aiming at the issue about the decreasing of the user service quality which was caused by much integrations of the resource, Wang, T et.al [28] solves the average optimization between the energy and the property by using the dynamic migration of the virtual machine. According to the platform of the SaaS, Zhao, G.Z et.al [29] proposed a sort of resource scheduling scheme to balance the energy consumption and the property. The technology of Job Scheduling Optimization Strategy (JSOS) appears as a new technology in the current low energy consumption design of the computer system, which leads to many new challenges and chances for the energy-saving scheduling of the progress. Using the feature of JSOS technology to dispatch the task can make the high performance cluster decrease the furthest cost of the energy in the whole system, in the situation of the consideration to the task performance. Sun, Z.Y et.al [30] proposed Heterogeneity-aware Loads Balancing (HLB) technology to balance distribution of the different task, which decrease the energy consumption overall and to set the constraint of the temperature. Based on the Task Distribution Layer Method (TDLM) technology, Wang, T et.al [31] promoted a sort of heuristic scheduling algorithm to reduce the executive energy consumption which was produced by the parallel task in the cluster environment. On the condition that this kind of algorithm does not influence the time of the whole tasks, the algorithm reduces the consumption of the energy by reducing the node voltage of the task. Liu, X.X et.al [32] also did the same research to put forward a kind of task scheduling algorithm based on the energy-aware in the integrated environment, which reduces the power consumption through controlling the proper level of the voltage. Aiming to the JSLM proposed a sort of task scheduling strategy of the energy-aware, which adjusts the needed voltage dynamically according to dividing the length of the subtask and minimize the energy consumption in the condition of satisfying the user deadline. Wu, Y.K et.al [33] promoted a sort of the energy saving system of the energy-aware, which judges the users are in the situation of activation or not by setting the dynamic reconfiguration models. Through turning on or turning off the application selectively to achieve the purpose of energy saving, this kind of system can acquire a good energy using rate. And at the same time, it can also provide different energy scheduling strategy. Aiming to the task scheduling problem in the heterogeneous computing environment, Liu, X.X et al [34] put forward an Energy- Conscious Scheduling (ECS) algorithm, which balances the task property and energy consumption by minimizing the value of Relative Superiority (RS) according to making the RS of the task property and the cost of the energy consumption as the optimization goal. In the basis of the optimization goal of the RS value, Sun, Z.Y et.al [35], combined the algorithm of the genes, purposed a meta-heuristic scheduling algorithm to optimize the property and the energy consumption of the tasks. Wang, J.F et.al [36] makes the energy consumption as the constraint condition and makes the property of the concurrent tasks as the optimizing goals. Sun, Z.Y et.al [37] put forward a scheduling framework of the heterogeneous energy awareness resource, which makes the heat balance get rid of the ‘heat point’ according to adjusting the voltage of the services dynamically. Li, Z.X et.al [38] put forward energy awareness task fusion technology to optimize the energy consumption of the cluster, which reduces the energy consumption of the system through the way of maximizing the method of using resources. Liu, X.X et.al [39] achieves the goal of optimizing the energy consumption by optimizing the free energy consumption and the running energy consumption effectively in the process of running the task. According to the parallel task scheduling, Shao, Y.L et.al [40] put forward an optimizing algorithm which concerns about the running property, the cost, the reliability, even the energy consumption and so on. However this algorithm needs much exploration in the features of the task and even the features of the RS, and, in the aspect of optimizing the energy, it also has much knowledge to explore.

In order to guarantee the fairness of the users work, the promoted work scheduling level method sets the dynamic priority for the work to guarantee the work priority scheduling method whose priority is high. When the source is limited, in the situation of not delaying the running of the original high priority work, the work priority of adjusting high priority dynamically will dispatch the work whose priority is low in advance, and at the same time it will save resource for the work whose original priority is high [41]. Once the work requirement whose source satisfies the high level priority stopped the work whose priority is low and restarted to patch the work whose priority is high, the mission will send the frame sequence to the unit by the way of making the unit as the source particle size basic unit, basing on the way of scene geometry to choose the key frame, using the feedback of the source occupying information of the key frame and the related feature of between the frames, combining the distributing strategy among the frames.

In Section 3.1, how can the authors guarantee the contention fairness when the data in the sensor-cloud computation apply for the resources?

We would like to thank the reviewer for the helpful suggestion. We would like to thank the reviewer for the comments. In sensor-cloud computation, we adopt the dynamically adjustable priority and contention mechanism to guarantee the fairness. That is, no job can postpone the execution of other jobs with higher priority. We adopt the ideology of appointment in order to guarantee the maximal data fusion rate. In this way, the un-fused data can obtain lower priority, and jobs with low resource requirement can be executed first. Therefore, we can pre-allocate computational resources for jobs with high priority and high resource requirement.

The explanation on Fig. 1 is over-simplified. Please supplement more details.

We would like to thank the reviewer for the helpful suggestion. Fig. 1(a) represents the initial state of the scheduling, when the system has idle resources, and the 5 jobs J1-J5 submitted from users are waiting to be scheduled by the job-managing system. The jobs J1, J3 and J5 have the highest priority. Therefore, J1, J3 and J5 are scheduled first. At this time, the system has 15 idle CPUs, which could fulfill the resource requirements of J1, J3 and J5. Therefore, data fusion is executed for the jobs J1, J3 and J5 in time T0. Since the system has no more idle resources, the resource requirement of job J2 and J4 cannot be met. Henceforth, J2 and J4 stay in the queue, waiting to be scheduled.

Fig. 1(b) represents that time T1 is the scheduling state. At this time, Job J5 has finished the rendering and there are 4 idle CPUs. According to the order of priority, J2 should be scheduled and allocated resources. However, J2 requires 8 CPUS, which cannot be met the system. For the job J4 with lower priority, its resource requirement can be met by the system. Therefore, in order to maximize the throughput and the resource efficiency, the job scheduling strategy schedules J4 in advance. Henceforth, the job J4 with lower priority is executed before J2 with higher priority.

Fig. 1(c) represents that time T2 is the scheduling state. At this time, the data fusion for J3 is finished, and the number of idle CPUs is 5 for the system. Therefore, the resource requirement of J2 still fails to be satisfied. However, the number of appointed CPUs is 4, and the total number of idle CPUs and appointed CPUs is 9, which could satisfy the requirement of J2. Meanwhile, the progress of J4 is smaller than 90%. Therefore, J4 is paused, and the CPUs occupied by J4 is now changed into idle state. Then, 8 idle CPUs are allocated to J2, and J4 is now in the queue waiting to be scheduled. At this time, since all the jobs in the queue require more than 1 CPUs, the system will generate 1 idle CPU.

Fig. 1(d) represents that time T3 is the scheduling state. At this time, the data fusion task for J2 is finished, and there are 9 idle CPUs in the system, which could satisfy the requirement of J4. Then, resources are allocated to J4 for the unfinished jobs. At this time, the scheduling queue is empty. The sensor-cloud computation system waits for users to submit jobs.

Please check the paper to avoid grammar mistakes and typos.

We would like to thank the reviewer for the helpful suggestion. We have checked the paper for the correctness of the grammar and typos.

Reviewer 3 Report

This paper proposes a hierarchical data job scheduling strategy Based on Intelligent Sensor-Cloud in Fog Computer (HDJS). HDJS dynamically adjusts the priority of the job to avoid job starvation and maximize the use of resources and uses the key frame to the resource occupied information, and distributes the frame sequence to the unit, and then combines the intra frame distribution strategy to balance the load between the nodes. Some improvements could be made as follows:

1.The innovative points could be highlighted and described in detail such as the first one.

2.Is it possible to combine Figure 6 (a) and (b) with (c) and (d) in one page?

3.The presentation of this paper could be improved.

Author Response

The innovative points could be highlighted and described in detail such as the first one.

We would like to thank the reviewer for the helpful suggestion. I have made relevant changes.

1) Setting the dynamic priority to guarantee the scheduling fairness.

2) The work that priority is high or whose requirement of source is big, and it also prevents situations of work starvation and reducing the source fragments.

3) Using the feedback of the source occupying information of the key frame and the related feature of between the frames and it combined the distributing strategy among the frames to render the work load between the nodes in balance.

Is it possible to combine Figure 6 (a) and (b) with (c) and (d) in one page?

We would like to thank the reviewer for the helpful suggestion. I have made relevant changes.

The presentation of this paper could be improved.

We would like to thank the reviewer for the helpful suggestion. We have checked the paper for the correctness of the grammar and typos.

Round 2

Reviewer 1 Report

All my suggestions were taken into account. I accordingly consider this paper suitable for publication.